



# The porosity effect on the mechanical properties of summer sea ice in the Arctic

Qingkai Wang, Yubo Liu, Peng Lu, Zhijun Li

State Key Laboratory of Coastal and Offshore Engineering, Dalian University of Technology, Dalian, 116024, China

*Correspondence to*: Qingkai Wang (wangqingkai@dlut.edu.cn)

**Abstract.** To investigate the mechanical properties of sea ice in the current summer Arctic, two ice blocks were lifted using ship crane during an Arctic expedition in the summer of 2021. Examination of ice crystal structure showed a granular ice layer at the top underlain by a columnar ice layer. Sea ice samples were then machined from the ice blocks for mechanical experiments performed in the laboratory. Three-point bending tests were conducted at ice temperatures of −12 to −3 ℃, and

uniaxial compressive strength tests were conducted at ice temperatures of −8 to −3 ℃ with a strain rate range of $10^{-6}$–$10^{-2}$ $s^{-1}$. The ice density and salinity of each sample were measured to determine brine and gas volume fraction as well as porosity. Results showed that sea ice flexural strength decreased with increasing porosity, but did not change with varying brine or gas volume fractions. A parameterization was proposed to relate sea ice flexural strength to porosity. The sea ice strain modulus was also independent on porosity and volume fractions of gas and brine. The uniaxial compressive strength decreased with

increasing porosity at both ductile and brittle strain rate regimes. Furthermore, three-dimensional surfaces were obtained to depict the sea ice uniaxial compressive strength varying with porosity and strain rate, based on which the transition strain rate from ductile to brittle behaviors was determined. It was found that the transition strain rate decreased with increasing porosity. Comparisons with previous studies on sea ice strength showed that the previously reported equations for sea ice flexural strength and strain modulus did not agree with the measured data. Compared with the strength calculated using early reported

sea ice porosity, the flexural strength and uniaxial compressive strength of summer Arctic sea ice decreased in recent decades, which probably brings positive feedback to the Arctic navigation.

## 1 Introduction

The mechanical properties of sea ice play important roles in the sea ice dynamic and engineering processes. In the recent years, the rate of Arctic warming was estimated to be faster than that previously reported (Rantanen et al., 2022), leading to an

accelerated melting of sea ice and probably weakened strength. With sea ice strength weakening, the ice cover is easier to be broken into floes under the action of waves, exposing more open leads and enhancing ocean–air influx. On the other hand, the ice force exerted on the ship and offshore construction in the ice-infested waters would be reduced, which causes excessive and uneconomic outcomes applying the design codes using sea ice mechanical properties derived from measurements conducted decades ago.

The uniaxial compressive strength, flexural strength, and strain modulus are commonly used sea ice mechanical properties in engineering applications. Therefore, these properties received considerable attention, especially in the past century when gas and oil exploration was developed rapidly in the Arctic regions (Kovacs, 1997; Timco and Frederking, 1990; Timco and O'Brien, 1994). While high quality investigations of sea ice mechanical properties have been sparse in the last few years (Bonath et al., 2019; Karulina et al., 2019; Skatulla et al., 2022; Strub-Klein and Høyland, 2012), although commercial and

tourist shipping are flourishing in the Arctic Ocean.

It is generally accepted that sea ice mechanics is largely dependent on its physical properties. Sea ice is a mixture consisting of pure ice, brine, gas, and other impurities. The phase composition of sea ice varies with ice temperature, and the ice strength varies with the fraction of solid ice. It is, therefore, rational to use the total porosity of sea ice (i.e., the sum of gas and brine





volume fractions) as the optimal parameter to parameterize sea ice strength. But not all sea ice mechanical properties have
been related to sea ice porosity. The relationship between sea ice uniaxial compressive strength and porosity has been
quantified in previous studies (Moslet, 2007; Timco and Frederking, 1990), whereas, the commonly used equations to calculate
sea ice flexural strength is still based on sea ice brine volume fraction (Timco and O'Brien, 1994). A restriction that exists in
the Timco and O'Brien's equation is that it is valid only for cold ice. For warm ice in the summer Arctic, the brine drainage
and meltwater flushing cause a less content of brine than gas (Wang et al., 2020), leading to an overestimated flexural strength
using Timco and O'Brien's equation. Case is more complicated for sea ice strain modulus since few studies have quantified
this mechanical property. The sea ice porosity has been adopted to estimate sea ice strain modulus in the engineering standards
of ISO19906 (2010), which, however, is replaced by sea ice brine volume fraction in the latest version standard of ISO 19906
(2019). Recent experiments conducted by Wang et al. (2022) showed that both brine volume fraction and porosity had no
statistically significant effect on sea ice strain modulus. Sea ice substructure is much more important, and the strain modulus
increased with increasing platelet spacing.

With the Arctic Ocean becoming more accessible in summer, marine activities have increased in this region, which leads to
higher demand for constructions purposely built for the summer window period. Therefore, understanding the mechanical
properties of summer sea ice is urgent. A question arises consequently is that whether the equations established on cold ice
years ago are appropriate for summer ice in the current Arctic. On one hand, sea ice physical properties change with sea ice
getting warmer during an annual cycle. The Multidisciplinary Drifting Observatory for the Study of Arctic Climate (MOSAiC)
expedition provided that sea ice salinity decreased and temperature increased due to the rapid warming and desalination of ice
in summer (Nicolaus et al., 2022). On the other hand, in response to the global warming, even for summer melting sea ice,
Wang et al. (2020) found that both density and salinity of sea ice in current years decreased than decades ago. Additionally,
latest studies indicated that the ice growth rate in the onset of freezing period becomes higher (Lei et al., 2022), and the
contribution of dynamic processes to the increase of ice thickness is enhanced (von Albedyll et al., 2022). The thermodynamic
and dynamic variations of Arctic sea ice in the early growing period would further change the sea ice microstructure. Therefore,
more studies are essential to update our knowledge on the Arctic sea ice mechanical properties since these properties have
probably changed, which could be insightful to the response of Arctic sea ice to global warming.

To investigate the mechanical properties of current summer Arctic sea ice and their dependences on sea ice porosity, ice blocks
were collected in the Central Arctic Ocean during ice melt season, which were stored for subsequent experiments on sea ice
physical and mechanical properties performed in the domestic laboratory. In this study, we presented the mechanical
experiment results of flexural strength, strain modulus, and uniaxial compressive strength of Arctic sea ice. Mathematical
equations were given to quantify the relationships between sea ice strength and porosity, which were further compared with
those reported in previous studies. The results will unify the physical parameter affecting sea ice mechanics to sea ice porosity.
These updated equations will also help design constructions aimed for working in Arctic summer.

**2 Field sampling and laboratory tests**

**2.1 Overview of field sampling**

During the Chinese National Arctic Research Expedition in 2021, two ice sites S1 (85.7 °N, 86.2 °E) and S2 (85.9 °N, 87.9 °
E) were set on the level ice along the ship cruise path in the Central Arctic Ocean on 9 August 2021 (UTC). When the icebreaker
stopped in the pack ice zone, broken ice turned over with whole cross section exposed and piled up beside the ship hull.
Therefore, a large ice block was lifted onto deck using ship crane at each ice site (Fig. 1). The detailed information of ice
blocks is listed in Table 1. The ice thickness was 130 cm at S1 site, and 160 cm at S2 site. The ice at both sites were covered
by 10 cm snow approximately, and the air temperatures during sampling were close to 0 ℃. Ice temperatures were measured
using an ice core extracted from the unbroken ice cover nearby the ship using a thermistor probe into holes drilled at 20 cm





intervals shortly after core extraction. The mean ice temperature was −0.6 ℃ at S1 site and −1.1 ℃ at S2 site, indicating the ice was melting. Based on the Arctic sea ice melt data issued by NASA's Goddard Earth Science Laboratories (https://earth.gsfc.nasa.gov/cryo/data/arctic-sea-icemelt, last access: 12 August 2022), the ice in the Central Arctic Ocean had experienced 40 day melt approximately at the time of sampling. Further visual check after the ice blocks were lifted onto deck showed no cracks on the appearance of ice, only the top of ice damaged partly due to interaction with ship hull. Immediately

after visual check, the two ice blocks were archived carefully using plastic bags to avoid sublimation, and stored at a surrounding temperature of −20 ℃ without solar radiation in a cold room. The ice blocks were finally shipped to a low temperature laboratory at home after a 2 month expedition for detailed studies of crystal texture and mechanical properties.

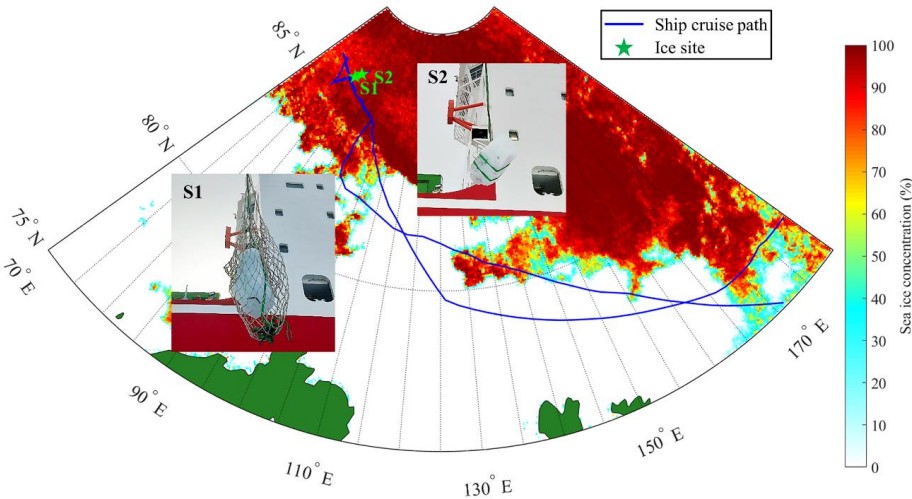

**Figure 1: Ice blocks extracted using ship crane at ice sites. The background is sea ice concentration on the day of sampling derived using AMSR2 data with a resolution of 6.25 km (https://www.seaice.uni-bremen, last access: 12 August 2022).**

**Table 1: The information of ice blocks.**

| Ice site | Ice thickness (cm) | Snow thickness (cm) | Air temperature (℃) | Ice temperature (℃) | Crystal structure |
|---|---|---|---|---|---|
| S1 | 130 | 10.7 | −0.1 | −0.6 | Top granular ice followed by |
| S2 | 160 | 11.9 | 0.2 | −1.1 | columnar ice |

   By preparing thin sections and observing under crossed polarized light, the texture characteristics of ice blocks at S1 and S2

sites were identified (Fig. 2). Both ice blocks showed a granular ice layer at the top underlain by columnar ice layer until the bottom, which is the typical texture structure of first-year ice. The vertical sections at top part of ice at S1 site were not shown because the thin sections melted after preparation due to increased surrounding temperature during cold laboratory defrosting. It could be judged that there was granular ice at the top 35 cm approximately of ice at S1 site based on the horizontal sections at this part. The granular ice layer at S2 site was at top 40 cm approximately. The granular ice was fine-grained with a size of

1–2 mm in diameter. The columnar ice was consecutive from the bottom of granular ice to the ice bottom, indicating that the ice grew in a calm water with no dynamic process.



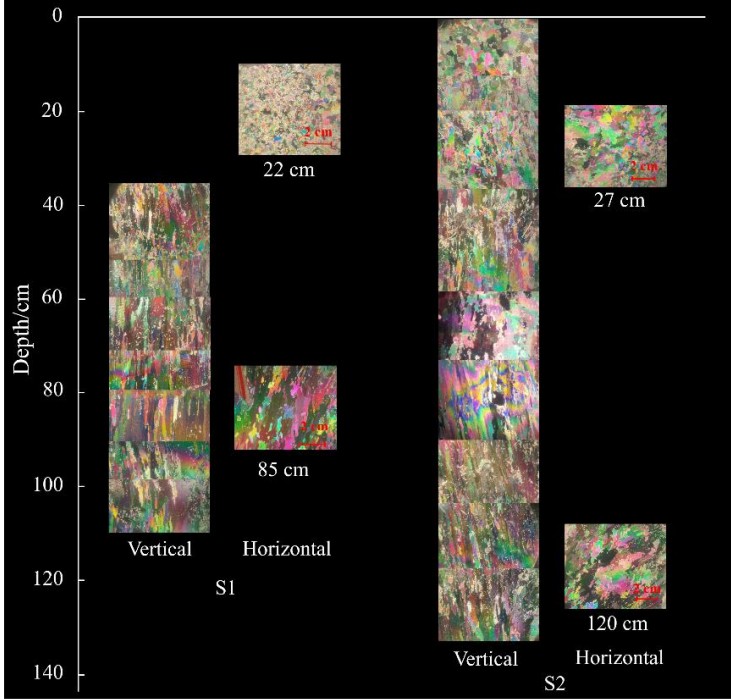

**Figure 2: The vertical stratigraphy of the ice texture profiles at S1 and S2 sites with typical horizontal thin sections selected. Vertical**
**sections at the top 35 cm at S1 site were not shown because the thin sections melt due to a mistake in operation.**

## 2.2 Laboratory mechanical experiments

### 2.2.1 Flexural test

The large ice blocks were machined into regular specimens in the cold laboratory before they were tested. Two types of sea ice mechanical tests were performed. The first type was three-point bending test to measure sea ice flexural strength and strain

modulus. Small ice beams with long axis horizontal to original ice surface were cut first from large blocks using chain saw, which were then processed carefully using band saw to make rectangular cross-section with dimensions of 7 ×7 cm. The long axis of the ice beam was finally cut to 55 cm using band saw at right-angles to the beam section. Because the ice samples were derived from melting sea ice, their salinities and brine contents were also small (see details in Sect. 3.1 and 3.3). To further investigate the porosity effect, more test ice temperatures were set as −12, −8, −5 and −3 ℃ to change the sea ice phase

composition. The ice samples after preparation were saved in a thermotank at the temperatures to be tested for at least 24 h to achieve phase equilibrium, and thus the sea ice porosities were ensured to be changed.

A small universal testing machine was used to perform the sea ice bending tests (Fig. 3a). The loaded plate of the machine was fitted with a stainless-steel column to give a line force on the midspan of the ice beam, which was supported by a frame with a span of 50 cm between two supports. The device was equipped with a force sensor of 3 kN capacity and ±1 % accuracy to

measure the force exerted on the beam midspan and a displacement sensor with ±1 % accuracy to measure the displacement of loaded plate, i.e., the beam midspan deflection. The stiffness of the machine was $2 \times 10^7 \, \mathrm{N \, m^{-1}}$; therefore, it was expected that there is a negligible difference between midspan deflection and loaded plate displacement, especially under the small force during ice bending test. Both force and displacement were recorded at frequencies of 50 Hz. The loading time of bending test was less than 30s (corresponding to a strain rate range of $10^{-5}$–$10^{-2} \, \mathrm{s^{-1}}$), located within the general loading time of sea ice

bending tests summarized by Timco and Weeks (2010).





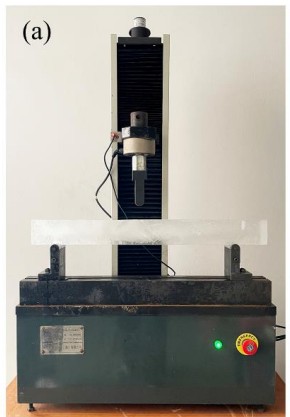
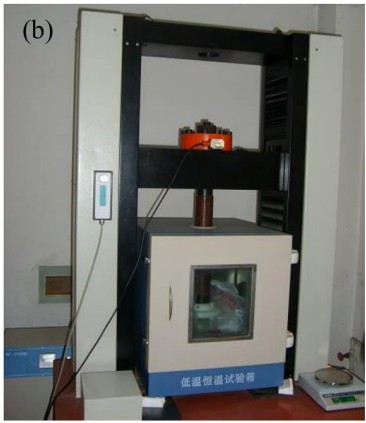

**Figure 3: Test machine for sea ice three-point bending test (a) and uniaxial compressive strength test (b).**

Before loading, the ice density was calculated using mass/volume method. The mass of each beam sample was measured using a balance (accuracy ±0.1 g), and the beam volume was calculated according to dimensions measured using a ruler (±1 mm)

for beam length and a caliper (±0.02 mm) for beam width and height. After failure, the broken ice was collected to melt for salinity measurement using a salinometer (±0.1 ppt). The brine and gas volume fractions as well as porosity of ice beam were then determined using sea ice temperature, salinity, and density according to Cox and Weeks (1983). A total of 44 ice beam samples were machined from the ice blocks, of which only four samples were granular ice, because the top parts of ice blocks were damaged and had insufficient space for preparation.

Thirty-eight ice beams with failure in the midspan were analyzed in this paper. The flexural strength and strain modulus were determined as Eq. (1) and (2):

$$\sigma_{\mathrm{f}} = \frac{3Fl}{2bh^2} \, , \tag{1}$$

$$E = \frac{Fl^3}{4bh^3\delta} \, , \tag{2}$$

where $\sigma_{\mathrm{f}}$ is ice flexural strength, $E$ is strain modulus, $F$ is load at ice failure, $l$ is span between supports, $b$ is sample width, $h$ is

sample height, and $\delta$ is the midspan deflection of beam. Equations (1) and (2) followed linear elasticity theory assuming that the ice beam is homogeneous and perfectly elastic, and were recommended by The IAHR Section on Ice Problems (Schwarz et al., 1981) and adopted by many other studies (Barrette, 2011; Karulina et al., 2019).

### 2.2.2 Uniaxial compressive strength test

The other type of sea ice mechanical test was uniaxial compressive strength test to measure sea ice uniaxial compressive

strength. The rough-cut ice cuboids were prepared first out of large ice block using a chain saw, which were then machined with care to section dimensions of 7 × 7 cm and length of 17.5 cm using the band saw. During the sample fashioning, both ends of the samples were planed using a spoke shave to keep them flat, and made vertical to sample long axis by checking with a square ruler. The ice samples were finally stored in a thermotank at required temperatures for tests (−8, −5, −3 °C) for at least 24 h.

The uniaxial compressive strength test was conducted using a large universal testing machine (Fig. 3b). Detailed description on the machine can be found in Wang et al. (2022). The machine is equipped with a force sensor of 100 kN capacity and ±0.5 % accuracy as well as a displacement sensor with ±2 μm accuracy. The loading speed can be controlled constant with an accuracy of ±0.5 % using a servo motor. Both force and displacement were recorded at frequencies of 50 Hz.





A total of 156 ice samples were prepared, of which 110 samples were columnar ice and 46 samples were granular ice. The
columnar ice samples were compressed in the directions vertical and horizontal to ice surface and the granular ice samples
were compressed in the direction horizontal to ice surface. The test strain rates were from $10^{-6}$ s$^{-1}$ to $10^{-2}$ s$^{-1}$. The mass and
dimensions of each sample were measured before compression to determine ice density using the same method as adopted in
the bending tests, and the fragments were collected to melt for salinity measurements after failure. The brine and gas volume
fractions as well as porosity of each ice sample were also determined according to Cox and Weeks (1983). The sea ice uniaxial
compressive strength was calculated as Eq. (3):

$$\sigma_{c} = \frac{F_{\max}}{bd}\tag{3}$$

where $\sigma_c$ is ice uniaxial compressive strength, $F_{\max}$ is maximum recorded force, and $d$ is sample length.

The nominal strain rate $\dot{\varepsilon}$ was applied here, which is defined as the ratio of machine loading speed to simple height. As the
failure load of uniaxial compressive samples is much higher than that of flexural sample, regarding the displacement of loaded
plate as the sample deformation may overestimate the true strain rate of ice sample (Timco and Frederking, 1984). The accurate
stiffness of our machine for uniaxial compressive strength test was not measured. Based on the machine capacity and material
construction (refer to Fig.3 in Wang et al. (2022)), it was expected to be rigid enough and brought a minor effect on the strain
rate.

### 2.2.3 Measurement uncertainty

As stated before, the sea ice strength was calculated according to failure load, deformation, and sample dimensions, therefore,
the uncertainties due to measurement error can be estimated with an error propagation analysis (see detailed calculation in
Appendix A). For bending test, the uncertainty of flexural strength is 1.1 % and that of strain modulus is 2.1 %; for compressive
strength test, the uncertainty of the uniaxial compressive strength is approximately 0.6 %. The uncertainty caused by the
measurement system is quite small than the inherent scatter of the mechanical test results. Considering the averages of
measured sea ice strength, the uncertainty of flexural strength was 8 kPa; the uncertainty of strain modulus was 0.1 GPa; the
uncertainty of uniaxial compressive strength is 0.01 MPa.

The gas and brine volume fractions as well as porosity of sea ice samples were calculated using ice temperature, salinity, and
density. The calculation would also involve uncertainties due to the measurement uncertainties of sea ice physical properties.
While the error propagation analysis could not be applied to determine the uncertainties of calculated sea ice phase composition
because sea ice temperature, salinity, and density were correlated. Very few studies have reported the gas volume fraction and
porosity of sea ice especially during bending tests in field because in situ density are difficult to obtain in the field.
Underestimation of ice density due to brine drainage is significant (Hutchings et al., 2015). Our tests were performed in the
laboratory, and the sea ice sample density was measured after they reached phase equilibrium. So the brine drainage would
not affect the density measurement, and the accuracy of density measurement could be guaranteed to an acceptable level. The
uncertainty of density measurement of ice samples was 0.2 % in the bending tests, and 0.7 % in the uniaxial compressive
strength tests according to error propagation analysis. Consequently, it was expected that the uncertainties of calculated sea
ice phase composition were also at an acceptable level.

## 3 Results

### 3.1 Flexural strength

A total of four granular samples were tested in the bending tests under −3 ℃ with an average of 369 kPa and standard deviation
of 72 kPa, which was lower than the flexural strength of columnar ice at the same test temperature (644 ± 195 kPa). The
flexural strength of our granular ice was weaker than that of columnar ice, because the porosity of granular ice (average 26.1





$\pm 6.3$ %) was higher than that of columnar ice (average 13.5 $\pm 4.5$ %). Given the small amount of granular ice samples, only the flexural strength of columnar ice was analyzed below.

The results of bending tests on columnar ice showed that the maximum flexural strength was 1149 kPa and the minimum value was 571 kPa. The porosity, brine volume fraction, and gas volume fraction of ice samples were 5–20 %, 0.2–2.5 %, and 3.5–18.6 %, respectively. In the previous studies, the flexural strength of cold ice was related to the brine volume fraction, and thus, the effect of brine volume on our ice strength was investigated first here. Since the brine volume fractions were small, to show the varying trend of flexural strength with brine volume fraction clearly, the fractions were expressed using square roots. To further reduce the effect of inherent scatter of sea ice mechanical tests, mean strength and standard deviation were determined taking the square root of brine volume fraction of 0.02 as a bin. The relationship between sea ice flexural strength and square root of brine volume fraction is depicted in Fig. 4a. The flexural strength decreased with increasing square root of brine volume fraction, and the power function performed better than other commonly used functions (e.g., linear, logarithmic, and power functions) with higher determination coefficient ($R^2$). However, even the highest $R^2$ was still less than 0.2, which was also not significant at a significance level ($p$) of 0.1.

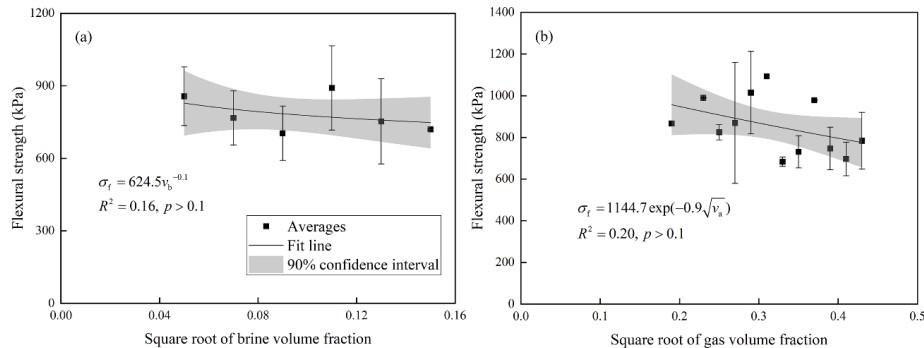

**Figure 4: The variations of sea ice flexural strength with square root of (a) brine volume fraction and (b) gas volume fraction. Also shown are the fit line and corresponding equation.**

It is noteworthy that the gas occupied much more space in ice samples than brine. Therefore, the effect of gas on the sea ice flexural strength cannot be neglected. For a better comparison with brine volume fraction, using the similar processing method as brine, the mean flexural strength and standard deviation were determined taking the square root of gas volume fraction of 0.02 as a bin. As shown in Fig. 4b, sea ice flexural strength decreased with increasing square root of gas volume fraction. The exponential function was the best regression form to depict the varying trend, but the relationship was also weak only with $R^2$ = 0.20 and not significant at $p$ = 0.1 level.

Sea ice porosity is the sum of brine and gas volume fractions, and the dependence of sea ice flexural strength on porosity was checked. The mean strength and standard deviation were determined taking the square root of porosity of 0.02 as a bin, and the relationship between sea ice flexural strength and square root of porosity is depicted in Fig. 5. Sea ice flexural strength decreased with increasing porosity. Regression analysis found that the exponential form (Eq. 4) performed best with higher $R^2$ = 0.32 out of other commonly used functions including linear, logarithmic, and power functions, which was significant at $p$ = 0.1 level.

$$\sigma_{\mathrm{f}} = 1193.2 \exp(-1.1\sqrt{v}) \quad (5\% < v < 20\%) \tag{4}$$

where $v$ is sea ice porosity.

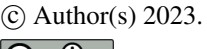



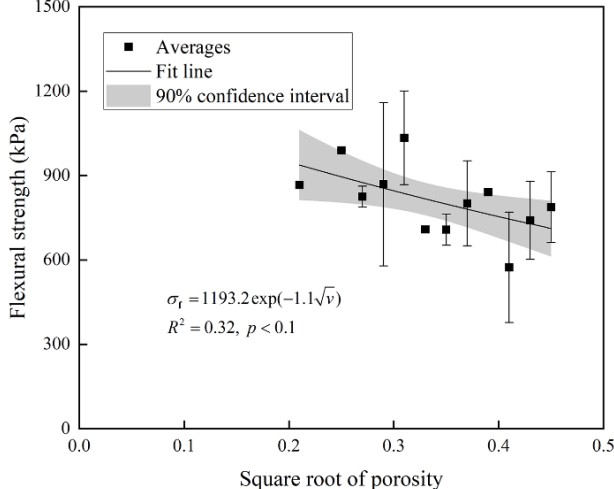

**Figure 5: The relationship between sea ice flexural strength and square root of porosity. Also shown are the fit line and corresponding equation.**

### 3.2 Strain modulus

230    The strain modulus of granular ice was 1.6–6.7 GPa with an average of 3.2 $\pm$ 2.3 GPa, which was similar to that of columnar ice of 1.6–8.4 GPa with an average of 3.1 $\pm$ 1.2 GPa. Considering the limited amount of granular ice samples, the strain modulus of granular ice was still excluded in the analysis below.

Using the similar processes as flexural strength, the strain modulus was averaged taking 0.02 of square root of porosity, brine volume fractions, and gas volume fraction as bins, respectively, to examine the dependences of sea ice strain modulus on the

235    ice phase composition. Sea ice strain modulus decreased with increasing square root of porosity (Fig. 6a). Further regression analysis gave the linear equation as the best fit form out of commonly used mathematical expressions, but the relationship was weak only with $R^2 = 0.15$ and not significant at 0.1 level, indicating that sea ice strain modulus was not dependent on sea ice porosity. The varying trends of sea ice strain modulus with brine and gas volume fractions are given in Figs. 6b and 6c, showing the strain modulus increased with increasing brine volume fraction and decreased with increasing gas volume fraction. The

240    best fit equations to depict the dependences of sea ice strain modulus on brine and gas volume fractions were further identified (Figs. 6b and c), while the relationships between strain modulus and brine as well as gas volume fractions were quite weak with $R^2 < 0.1$ (even approaching 0 for brine volume fraction) and not significant at 0.1 level.



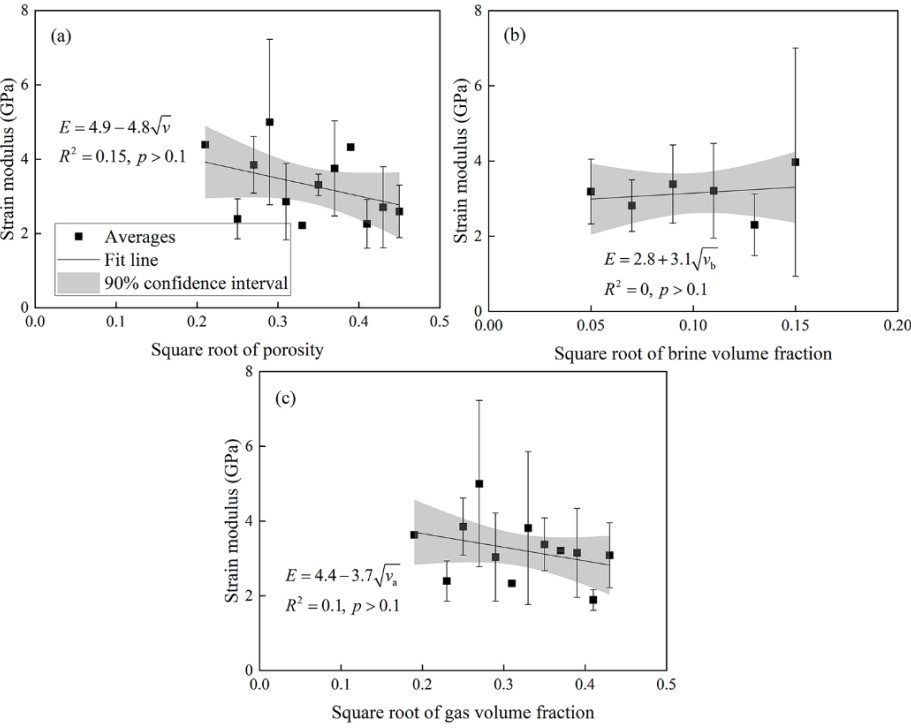

**Figure 6: The variations of sea ice strain modulus with square root of (a) porosity, (b) brine volume fraction, and (c) gas volume fraction.**

### 3.3 Uniaxial compressive strength

The uniaxial compressive strength of sea ice can be divided into ductile and brittle regimes, respectively, according to failure modes and stress-strain curves of samples. Detailed descriptions on the differences between ductile and brittle failures of ice samples have been reported in many studies, e.g., Wang et al. (2022), and thus were given briefly here. Ductile failure occurred at low strain rates, and the stress concentration in the ice samples were relaxed by local cracks development, which led to a final deformation without abrupt collapse. Therefore, stress decreased gently after reaching peak in the stress-strain curve. At high strain rates, local cracks and deformation were not sufficient to relax the stress concentration, which caused cracks penetrating the sample immediately after the force reached peak, and stress dropped abruptly with the sample collapsing.

The brine and gas volume fractions were separated first to check their respective influences on uniaxial compressive strength. The brine and gas volume fractions of compressive samples were 0–3.8 % and 1.8–36.9 %, respectively. Using similar processes as analyzing flexural strength, the uniaxial compressive strength of vertically loaded columnar samples, horizontally loaded columnar samples, and granular samples were averaged taking the square root of brine and gas volume fractions of 0.02 as a bin. Regression analysis showed that there were no significant dependences of uniaxial compressive on brine and gas volume fractions ($R^2 < 0.2$ and $p > 0.1$).

It is generally accepted that sea ice porosity is the primary factor affecting sea ice uniaxial compressive strength. The sea ice porosity was 3.3–24.7 %, 9.0–21.9 %, and 9.8–36.9 % for vertically loaded columnar samples, horizontally loaded columnar samples, and granular samples, respectively. Figure 7 shows the variations of uniaxial compressive strength with sea ice porosity, in which the porosity was not expressed using square roots as it did for flexural strength, considering that the ranges were similar approximately between porosity and square root of porosity. The mean strength and standard deviation were determined taking 0.05 of porosity as a bin in Fig. 7. The uniaxial compressive strength decreased with increasing porosity, and further regression analysis showed that the varying trends could be described using the power law for all three types of



tests with $R^2 > 0.8$ and at $p < 0.1$ level at least. It was also noted that the brittle strength was no less than ductile strength for an ice sample with the same porosity, but the ratio of brittle to ductile strength decreased with increasing porosity and

approached 1 finally. The trend was clear for horizontally loaded columnar ice. The ratio was 1.5 approximately at the porosity of 10–15 %, and decreased to 1.2 approximately with porosity above 15 %. For vertically loaded columnar ice, judging from the trend lines, the ratio was higher than 1.2 for porosity of 10–15 %, and as porosity was higher than 15 %, the two curves coincided nearly. For granular ice, both the measured data and trend lines for brittle and ductile strength were close.

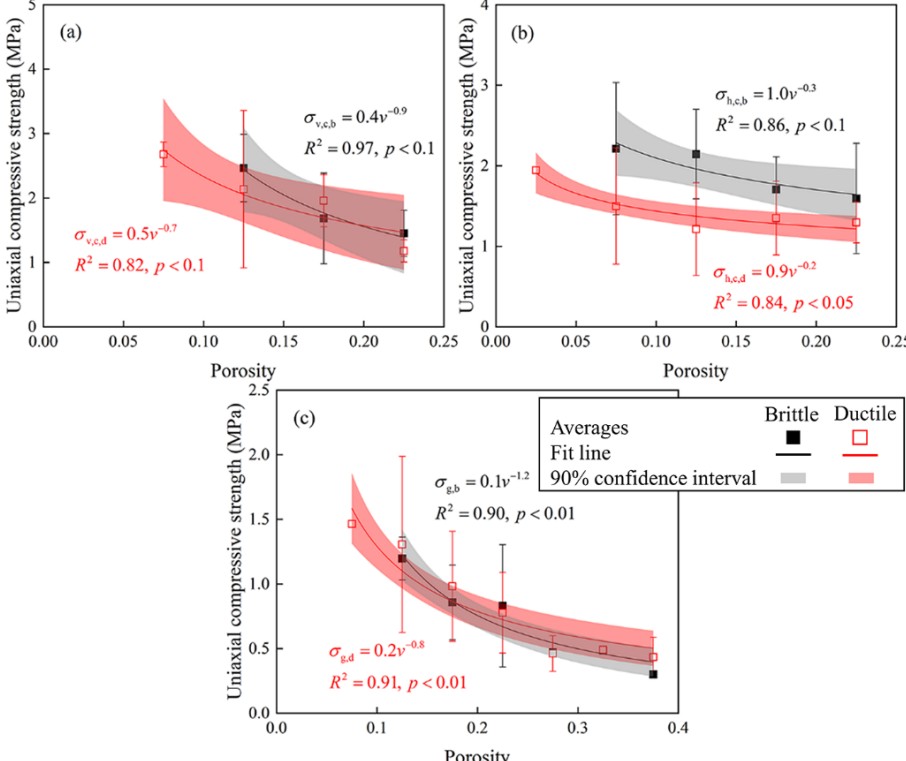

**Figure 7: The variations of sea ice uniaxial compressive strength with porosity for (a) vertically loaded columnar samples, (b) horizontally loaded columnar samples, and (c) granular samples. The subscripts v, h, c, g, b and d represent vertical loading, horizontal loading, columnar ice, granular ice, brittle regime, and ductile regime, respectively.**

It was noteworthy that the effect of strain rate on the uniaxial compressive strength was not separated in the Fig. 7. Therefore,

the standard deviation was high. It is well known that sea ice uniaxial compressive strength is largely affected by strain rate. In the ductile strain rate regime, the uniaxial compressive strength increases with increased strain rate following a power law (Moslet, 2007); and when ice fails in a brittle way, our previous studies found that the power law could also be used to describe the varying trend of uniaxial compressive strength with strain rate (Wang et al., 2022). Consequently, both sea ice porosity and strain rate were required to describe the variations of sea ice uniaxial compressive strength in ductile and brittle regimes.

The three-dimensional surfaces shown in Fig. 8 exhibited the varying trends of uniaxial compressive strength, in which Eq. (5) was adopted using two-parameter regression analysis, given the respective mathematical forms of uniaxial compressive strength with porosity and strain rate.

$$\sigma_c = A\dot{\varepsilon}^B v^C \tag{5}$$

where $\dot{\varepsilon}$ is strain rate, and $A$, $B$, and $C$ are fitting coefficients listed in Table 2.



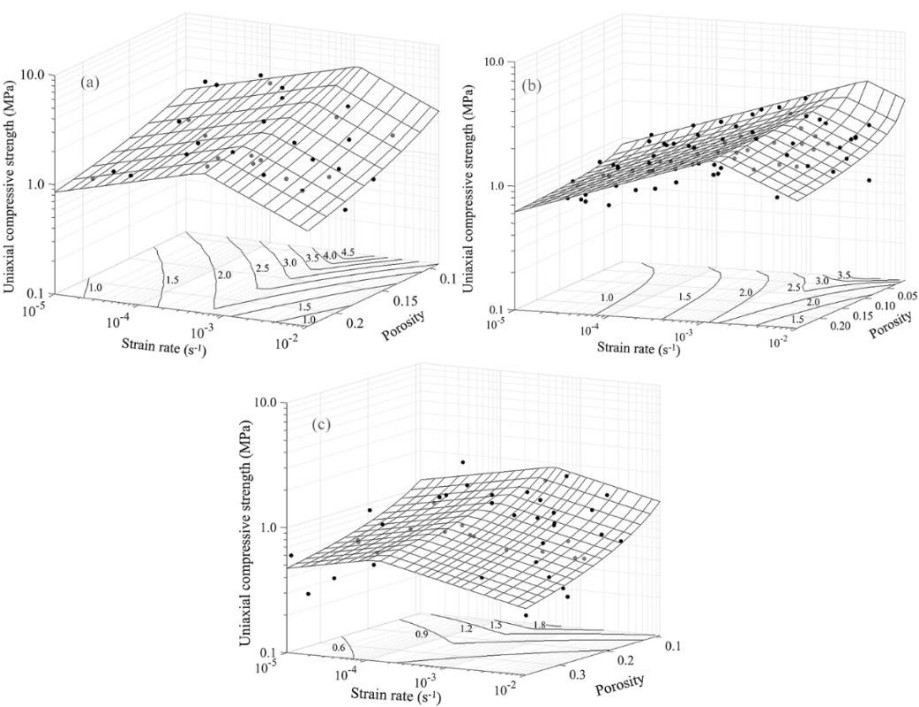

**Figure 8: Three-dimensional surfaces of sea ice uniaxial compressive strength varying with porosity and strain rate for (a) vertically loaded columnar samples, (b) horizontally loaded columnar samples, and (c) granular samples.**

**Table 2: The fitting coefficients of Eq. (5).**

| Sample | Ductile regime | | | | Brittle regime | | | |
|---|---|---|---|---|---|---|---|---|
| | $A$ | $B$ | $C$ | $R^2$ | $A$ | $B$ | $C$ | $R^2$ |
| Vertically loaded columnar | 2.27 | 0.20 | −0.91 | $0.51^\alpha$ | 0.02 | −0.33 | −1.29 | $0.74^\alpha$ |
| Horizontally loaded columnar | 9.18 | 0.25 | −0.11 | $0.60^\alpha$ | 0.15 | −0.33 | −0.31 | $0.43^\alpha$ |
| Granular | 1.02 | 0.12 | −0.63 | $0.48^\alpha$ | 0.07 | −0.15 | −0.88 | $0.42^\beta$ |

$^\alpha$ and $^\beta$ represent significance levels of 0.01 and 0.05, respectively.


It also could be found from Fig. 8 that there was a different strain rate beyond which the uniaxial compressive strength transits from ductile to brittle regime at different porosities. The transition strain rates were $4.0\times10^{-4}$–$1.0\times10^{-3}$ s$^{-1}$ for vertically loaded columnar samples, $1.0\times10^{-3}$–$3.0\times10^{-3}$ s$^{-1}$ for horizontally loaded columnar samples, and $1.0\times10^{-4}$–$4.0\times10^{-4}$ s$^{-1}$ for granular samples. Moreover, the transition strain rate decreased with increasing sea ice porosity (Fig. 9). The transition from ductile to

brittle behavior of sea ice can be regarded as a competition between stress relaxation and stress build-up (Schulson, 2001). The stress build-up occurs at crack tips. Brine inclusions and gas bubbles in sea ice work as pre-cracks facilitating stress concentration at crack tip. Therefore, with sea ice porosity increasing, stress concentration can be triggered at a lower strain rate. In addition, the decreasing trend of transition strain rate with sea ice porosity can be modelled using a power law, and Fig. 9 gave the fitting results.

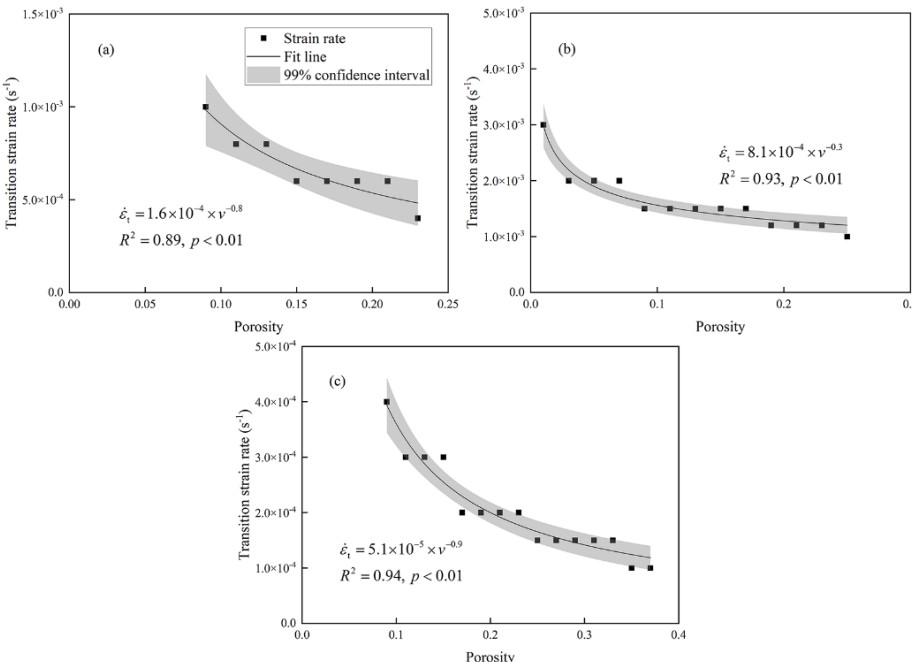


**Figure 9: The variations of transition strain rate with sea ice porosity for (a) vertically loaded columnar samples, (b) horizontally loaded columnar samples, and (c) granular samples. Also shown are the best-fit lines and equations.**

## 4 Discussion

### 4.1 Comparisons with previous studies

#### 4.1.1 Uniaxial compressive strength

Several studies have been conducted to investigate the sea ice uniaxial compressive strength, in which empirical equations were proposed to relate sea ice uniaxial compressive strength to porosity and strain rate (Timco and Frederking, 1990; Wang et al., 2022). The previously reported equations are given in Table 3. Taking the measured strength of 0.5 MPa as a bin, the average and standard deviation of estimated strength were obtained, and Fig. 10 compared the sea ice uniaxial compressive

strength measured in this paper with those estimated using empirical equations given by Timco and Frederking (1990) and Wang et al. (2022). It is noteworthy that the applicable strain rate is $1.0 \times 10^{-7}$–$2.0 \times 10^{-4}$ s$^{-1}$ in Timco and Frederking (i.e., ductile regime in their study) and $1.0 \times 10^{-6}$–$1.0 \times 10^{-2}$ s$^{-1}$ in Wang et al. (2022). Results indicated that the measured and estimated uniaxial compressive strength showed a good agreement with correlation coefficients higher than 0.90 for horizontally loaded columnar ice. While for vertically loaded columnar ice, both estimations by Timco and Frederking (1990)

and Wang et al. (2022) overestimated the measured strength. No equation was proposed in Wang et al. (2022) to estimate the uniaxial compressive strength of granular ice, and only Timco and Fredeking (1990) was applied for comparison. A good agreement was also shown between measured and estimated uniaxial compressive strength of granular ice.


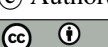



**Table 3: The previously reported equations for sea ice uniaxial compressive strength, flexural strength, and strain modulus. The**
**meaning of the subscripts can be referred to Fig. 7.**

| Variables | References | Equations |
|---|---|---|
| Uniaxial compressive strength | Timco and Frederking (1990) | $\sigma_c = A\dot{\varepsilon}^B(1-\sqrt{\dfrac{v}{C}})\ (1.0\times10^{-7} \leq \dot{\varepsilon} \leq 2.0\times10^{-4}\,\mathrm{s}^{-1})$<br>$A_{v,c}=160,\ B_{v,c}=0.22,\ C_{v,c}=200$<br>$A_{h,c}=37,\ B_{h,c}=0.22,\ C_{h,c}=270$<br>$A_g=49,\ B_g=0.22,\ C_g=280$ |
| | Wang et al. (2022) | $\sigma_c = A\dot{\varepsilon}^B v^C\ (1.0\times10^{-6} \leq \dot{\varepsilon} \leq 1.0\times10^{-2}\,\mathrm{s}^{-1})$<br>$A_{v,c,d}=325.75,\ B_{v,c,d}=0.13,\ C_{v,c,d}=-0.7$<br>$A_{v,c,b}=7.88,\ B_{v,c,b}=-0.26,\ C_{v,c,b}=-0.48$<br>$A_{h,c,d}=153.87,\ B_{h,c,d}=0.25,\ C_{h,c,d}=-0.58$<br>$A_{h,c,b}=2.97,\ B_{h,c,b}=-0.34,\ C_{h,c,b}=-0.52$ |
| Flexural strength | Karulina et al. (2018) | $\sigma_f = 526.6\exp(-2.804\sqrt{v_b})\,(\sqrt{v_b} \leq 0.5)$ |
| | Timco and O'Brien (1994) | $\sigma_f = 1760\exp(-5.88\sqrt{v_b})\,(\sqrt{v_b} \leq 0.5)$ |
| | Wang et al. (2022) | $\sigma_f = 1859.1\exp(-3.51\sqrt{v})$ |
| Strain modulus | ISO19906 (2019) | $E = 5.31 - 0.436\sqrt{v_b}$ |
| | Karulina et al. (2018) | $E = 3.1031\exp(-3.385\sqrt{v_b})\,(\sqrt{v_b} \leq 0.5)$ |

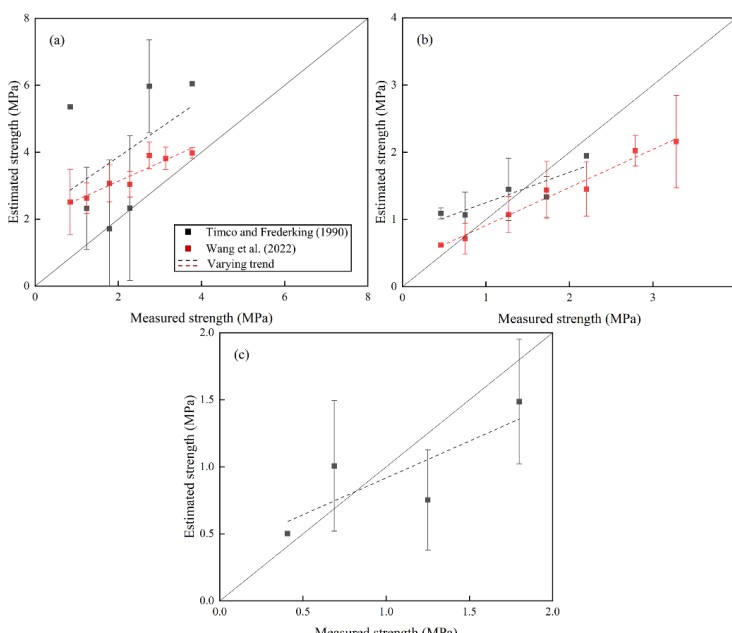

**Figure 10: Comparisons between measured uniaxial compressive strength with estimated strength using previously reported**
**equations for (a) vertically loaded columnar samples, (b) horizontally loaded columnar samples, and (c) granular samples.**

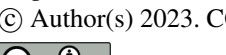



### 4.1.2 Flexural strength and strain modulus

Comparisons were also conducted for our measured flexural strength with those estimated using previously established equations in Karulina et al. (2019), Timco and O'Brien (1994), and Wang et al. (2022). Table 3 gives the detailed formulae. The former two equations relate sea ice flexural strength to brine volume fraction, and the latter is established based on sea ice porosity. Comparison results are shown in Fig. 11 by taking the measured strength of 100 kPa as a bin and determining the average and standard deviation of estimated strength. The estimated strength using Karulina et al. (2019) and Timco and O'Brien (1994) varied slightly. Because our ice samples were derived from melting ice, and the square root of brine volume fraction varied in a narrow range of 0.05–0.15, leading to minimal fluctuations of calculated flexural strength. Furthermore, the estimations by Karulina et al. (2019) and Timco and O'Brien (1994) behaved differently. A total of 939 reported measurements on flexural strength of sea ice in both polar and temperate climates using both full-scale cantilever beam test and small size simple test were compiled in Timco and O'Brien (1994). Therefore, the equation was representative and widely adopted. The overestimation of Timco and O'Brien (1994) than our measured strength confirmed that flexural strength more accurately depended upon the porosity, especially for warm sea ice. Karulina et al. (2019) underestimated our measurements. This was because their experiments were performed by full-scale cantilever beam tests. More potential weakness contained in the large beam and stress concentrations at the root of the beam caused lower flexural strength. In contrast, estimations using Wang et al. (2022) performed better than the other two equations. The underestimation could be owed to the differences of crystal structures.

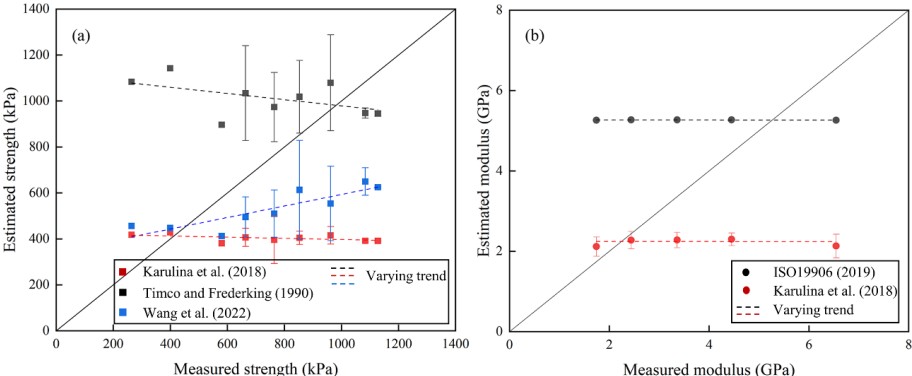

**Figure 11: Comparisons between measured (a) flexural strength and (b) strain modulus with estimations using previously reported equations. The error bars of estimated modulus using ISO19906 (2019) were quite short because of small standard deviation.**

It is interesting to note that the commonly used parameterization of flexural strength for cold growing ice using sea ice brine volume fraction is not appropriate for warm melting ice (also can be found in Figs. 4 and 5). Brine and gas are important inclusions in sea ice, and both of them cannot bear load. For cold growing ice, brine occupies more space in ice than gas, and sea ice flexural strength has been parameterized only using brine volume fraction, while, for warm melting ice, brine escapes due to drainage and meltwater flush, leading to minor space proportion than gas. Therefore, the gas effect on strength cannot be ignored. Combined with Figs. 4 and 5, neither brine volume fraction nor gas volume fraction can be used solely to parameterize sea ice flexural strength, indicating that sea ice porosity should be the controlling factor for warm sea ice. From a physical point of view, it is also rational to use sea ice porosity to parameterize the flexural strength of cold ice; but very few studies reported ice density during the flexural tests conducted on cold ice. If such cases are completed in future, a single formula could be obtained to estimate flexural strength for both cold and warm ice using sea ice porosity. Admittedly, as stated before, the range of square root of brine volume fraction of our ice samples was small, which is a possible reason why our flexural strength was not dependent on brine volume fraction.



Figure 11 also compares the measured strain modulus with estimated modulus using equations proposed in ISO19906 (2019) and Karulina et al. (2019), which are both established according to sea ice brine volume fraction (Table 3). Taking the measured

modulus of 1.0 GPa as a bin, the average and standard deviation of estimated modulus were obtained. Because of the narrow range of square root of brine volume fraction, the estimated modulus kept constant approximately. The estimation by ISO19906 (2019) is conservative with higher value, and Karulina et al. (2019) underestimated our measured data.

**4.2 Potential variation of summer Arctic sea ice strength**

It has been widely known that Arctic sea ice is undergoing a dramatic variation due to global warming, such as reductions in

sea ice thickness, volume, and multiyear ice coverage (Kwok, 2018). Recent studies also found substantial changes in the physical properties (e.g., ice salinity and density) of current summer Arctic sea ice than decades ago (Wang et al., 2020). Sea ice mechanical properties are largely controlled by its physical properties, hence, changes of sea ice strength may have already occurred in the summer Arctic.

The changes of sea ice density and salinity produce variations of sea ice porosity, which further changes sea ice mechanical

strength. Consequently, based on the quantitative relationship between sea ice strength and porosity, if the variation of sea ice porosity in the recent Arctic summers is obtained, the varying trend of Arctic summer sea ice strength could be then evaluated. However, very few studies have reported the sea ice porosity or given a complete dataset of sea ice temperature, salinity, and density. Table 4 collected the sea ice porosities in the Arctic summers published in previous reports. Overgaard et al. (1983) measured the profiles of temperature, salinity, and density of multiyear ice cores sampled in the European Arctic in 1978/79

summers. The data of ice cores were used here to calculate sea ice porosity. Brine and gas volume fractions varying with ice depth were given by Eicken et al. (1995), which were computed from temperature, salinity, and density profiles of multiyear ice cores collected in 1991 summer in the Eurasian sector of the Arctic Ocean. The mean sea ice porosity is determined in Table 4. Wang et al. (2020) compiled profiles of temperature, salinity, density of first-year and multiyear ice cores taken in the Pacific sector of the Arctic in summers of 2008–2016, according to which the average sea ice porosities were determined

in individual year in Table 4.

**Table 4: The reported sea ice porosity in the summer Arctic.**

| First-year ice | | | Multiyear ice | | |
|---|---|---|---|---|---|
| Date | porosity | Reference | Date | porosity | Reference |
| Aug. 2008 | 25.3 % | | Jul.–Aug. 1979 | 16.2 % | Overgaard et al. (1983) |
| Aug. 2010 | 20.5 % | | Aug.–Sep. 1991 | 16.0 % | Eicken et al. (1995) |
| Aug.–Sep. 2012 | 27.5 % | Wang et al. (2020) | Aug. 2008 | 20.6 % | |
| Aug. 2014 | 32.5 % | | Aug.–Sep. 2012 | 19.3 % | Wang et al. (2020) |
| Aug. 2016 | 28.5 % | | Aug. 2014 | 27.5 % | |
| | | | Aug. 2016 | 23.5 % | |

The calculated strength of first-year and multiyear ice are shown in Fig. 12, in which the uniaxial compressive strength was

taken as the average of ductile and brittle strength. The flexural and uniaxial compressive strength of sea ice after the year 2000 were determined using Eqs. (4) and (5). Because the previously reported equation of sea ice flexural strength was derived from cold ice, we adopted Eq. (4) to calculate the flexural strength of summer sea ice before the year 2000. For the uniaxial compressive strength of sea ice decades ago, we intended to calculate it using the equations derived from ice at about the same era. However, to our best knowledge, no formulae were found to quantify the relationship between sea ice uniaxial compressive

and porosity in previous studies, and strain rate always exists in the equations. Considering the relatively good agreements between estimated uniaxial compressive strength using the equation reported in the last century and our measured data (Fig.



10), we adopted the quantitative relationships shown in Fig. 7 to determine the uniaxial compressive strength of sea ice before the year 2000 based on sea ice porosity.

It could be seen that the strength of both first-year and multiyear ice in the summer Arctic decreased yearly. For first-year ice,
the calculated flexural strength decreased from 686 kPa to 663 kPa with a declining rate of 6 kPa per year. The flexural strength of multiyear ice was a little higher than that of first-year ice, and the strength decreases from 766 kPa to 700 kPa with a declining rate of 2 kPa per year. The uniaxial compressive strength of multiyear ice was a little more than that of first-year ice, while the declining trends were similar for the uniaxial compressive between first-year and multiyear ice. The declining trend was 0.04 MPa per year, 0.01 MPa per year, and 0.01 MPa per year for vertically loaded columnar ice, horizontally loaded
columnar ice, and granular ice, respectively.

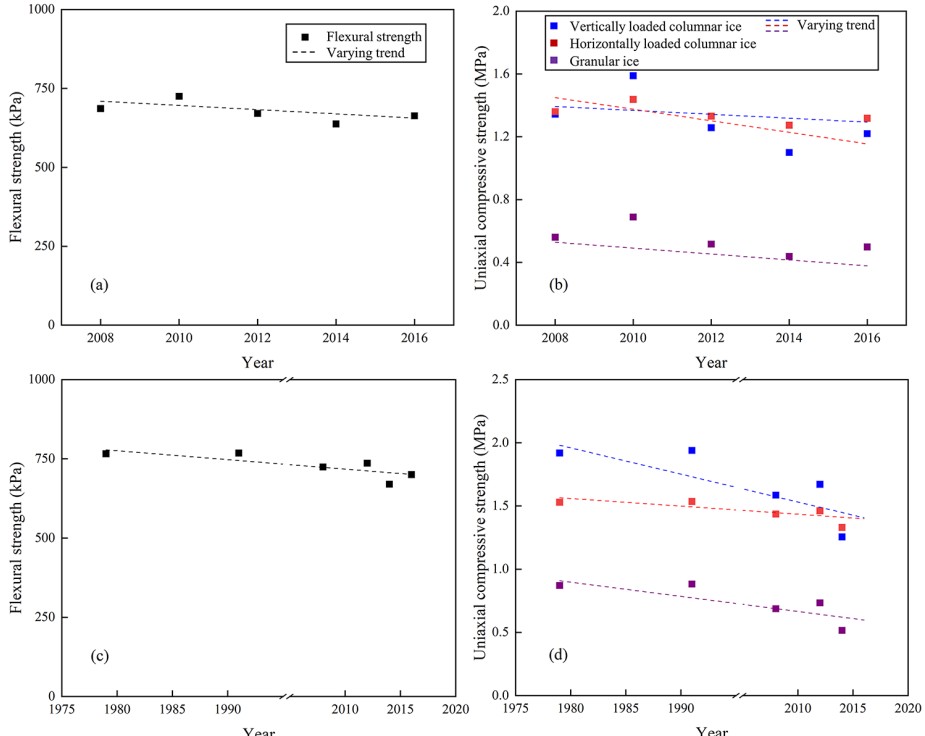

**Figure 12: Interannual variability of flexural and uniaxial compressive for (a) (b) first year and (c) (d) multiyear ice.**

Global warming has weakened the strength of Arctic summer sea ice, which will bring potential influences to Arctic navigation.
Sea ice flexural strength is a critical parameter affecting the ice resistance for ships in ice-covered waters. There is a widely used semi-empirical equation proposed by Lindqvist (1989) to determine level ice resistance on an ice-going ship according to sea ice flexural strength. Taking Chinese RV *Xuelong* as an example, the effect of decreased sea ice flexural strength on ice resistance on ship is evaluated. The RV *Xuelong* has completed several ice navigations in summer Arctic, which is capable of continuously navigating water with 1.2 m thick level ice covered by 0.2 m thick snow at a velocity of 1.5 knot ($\approx 0.77$ m s$^{-1}$)
in the polar regions. The parameters of RV *Xuelong* and Arctic summer sea ice properties involved in calculation of Lindqvist's equations are listed in Table 5. The ice thickness and ship velocity were taken based on the ship icebreaking ability. The ice friction coefficient was set as 0.05 following industry standard (Swedish Transport Agency, 2019). Ice density was 808.5 kg m$^{-3}$ taking the expected value of Beta distribution derived from probabilistic analysis of filed measurements (Wang et al., 2021). Sea ice flexural strength was set to 660–760 kPa according to the calculated strength of first-year and multiyear ice.
Calculation results showed that the ice resistance decreases by 8.5 % from 1.47 MN to 1.35 MN with sea ice flexural strength





decreasing from 760 kPa to 660 kPa. The decreased ice resistance implied that parts of the fuel costs were saved, facilitating a more economical Arctic navigation in current and future summers. The transport efficiency analysis conducted in von Bock und Polach et al. (2015) for ice-going vessels in Arctic voyages also showed an increased efficiency with less sea ice flexural strength.


**Table 5: The parameters of ship and sea ice used to calculate ice resistance.**

| Item | Value | Item | Value |
|---|---|---|---|
| Length (m) | 147 | Ice thickness (m) | 1.2 |
| Breadth (m) | 23 | Ice density (kg m$^{-3}$) | 808.5 |
| Draught (m) | 8 | Ice friction coefficient | 0.05 |
| Waterline entrance angle (°) | 20 | Ship velocity (m s$^{-1}$) | 0.77 |
| Stem angle (°) | 24 | Ice flexural strength (kPa) | 660–760 |

On the other hand, from the view of natural process, combined with thinner ice thickness and weaker ice strength in the summer Arctic, ice cover is more likely to be broken under the force of wind and wave, which is a possible reason leading to more lead

area fraction in the summer. The phenomenon has also been represented by sea ice models (Wang et al., 2016), which showed the summer lead area fraction has increased by about 60–80 % during the past three decades in the Arctic Ocean. As a result, atmosphere–sea interaction was strengthened, leaving more solar heat absorbed by upper ocean through open leads. The ice thickness thinning may be accelerated with increasing open leads through the sea ice–albedo positive feedback.

**5 Conclusions**

Arctic sea ice blocks were taken during the Chinese National Arctic Research Expedition in 2021 summer for mechanical experiments. Three-point bending tests and uniaxial compressive strength tests were carried out to measure the sea ice flexural strength, strain modulus, and uniaxial compressive strength. The porosity of each ice sample was determined according to the measurements of ice temperature, salinity, and density.

Both flexural strength and uniaxial compressive strength of summer Arctic sea ice were dependent on sea ice porosity, and

showed declining trends with increasing porosity. The sea ice flexural strength was independent on brine volume fraction, which was opposite to the phenomenon derived from cold ice. Therefore, an equation was established to relate sea ice flexural strength to porosity for summer Arctic sea ice rather than using brine volume fraction (Eq. 4). The uniaxial compressive strength showed different dependences on sea ice porosity at ductile and brittle strain rate regimes. It is clear to use three-dimensional surface to depict the variation of sea ice uniaxial compressive strength with strain rate and porosity. Furthermore,

the transition strain rate from ductile to brittle behaviors of sea ice could be determined from the surface, and the transition strain rate decreased with increasing porosity. Unlike sea ice flexural strength and uniaxial compressive strength, sea ice porosity, brine volume fraction, and gas volume fraction had no statistically significant effects on sea ice strain modulus.

The previously reported equations for sea ice flexural strength and strain modulus using brine volume fraction were not appropriate for estimating the strength of Arctic sea ice in current summers. By contrast, the previously reported equations for

sea ice uniaxial compressive strength using porosity performed better. Overestimation was obtained only for estimated uniaxial compressive strength of vertically loaded columnar ice than measured data, and for horizontally loaded columnar ice and granular ice, good agreements were shown between estimated and measured strengths. The comparisons also indicated that sea ice porosity was a better parameter than brine volume fraction to quantify the dependence of sea ice strength on its physical properties.



Using the dataset of physical properties of summer Arctic sea ice published in previous and recent reports, the sea ice flexural
strength and uniaxial compressive strength were calculated to explore the annual variation of summer Arctic sea ice strength
in recent 40 years. It was found that both flexural strength and uniaxial compressive strength in the summer Arctic decreased,
with a rate of 2 kPa per year and 0.01 MPa at least, respectively, for first-year and multiyear ice. The weakened strength of
summer Arctic further caused positive effect to Arctic navigation and enhanced atmosphere–sea interaction, which provided
a new perspective to explain global warming affecting the natural processes and industry activities in the Arctic Ocean.

**Appendix A: Calculation process of measurement uncertainty**

The sea ice flexural strength is calculated according to measurements of failure load and sample dimensions (see Eq. 1).
Uncertainty of sea ice flexural strength due to measurement error can be estimated with an error propagation analysis, which
is given by Eqs. (A1) and (A2):

$$\frac{\Delta\sigma_{\mathrm{f}}}{\sigma_{\mathrm{f}}} = \left|\frac{\partial\ln\sigma_{\mathrm{f}}}{\partial F}\right|\Delta F + \left|\frac{\partial\ln\sigma_{\mathrm{f}}}{\partial b}\right|\Delta b + \left|\frac{\partial\ln\sigma_{\mathrm{f}}}{\partial h}\right|\Delta h \ , \tag{A1}$$

$$\frac{\Delta\sigma_{\mathrm{f}}}{\sigma_{\mathrm{f}}} = \frac{\Delta F}{F} + \frac{\Delta b}{b} + \frac{2\Delta h}{h} \ , \tag{A2}$$

where $\Delta(\cdot)$ is the errors of corresponding parameters, $\sigma_{\mathrm{f}}$ is sea ice flexural strength, $F$ is load at ice failure, $b$ is beam width,
and $h$ is beam height. $\Delta F/F = 1$ % according to the performance of force sensor. $\Delta b = \Delta h = 0.02$ mm. $b = 68.3$ mm and $h =$
67.7 mm taking the average sample dimensions. Therefore, $\Delta\sigma_{\mathrm{f}}/\sigma_{\mathrm{f}}$ is 1.1 %.

For sea ice strain modulus, it is calculated based on measurements of failure load, sample dimensions, and midspan deflection
of beam (see Eq. 2). Uncertainty of sea ice strain modulus is given by Eqs. (A3) and (A4):

$$\frac{\Delta E}{E} = \left|\frac{\partial\ln E}{\partial F}\right|\Delta F + \left|\frac{\partial\ln E}{\partial b}\right|\Delta b + \left|\frac{\partial\ln E}{\partial h}\right|\Delta h + \left|\frac{\partial\ln E}{\partial\delta}\right|\Delta\delta \ , \tag{A3}$$

$$\frac{\Delta E}{E} = \frac{\Delta F}{F} + \frac{\Delta b}{b} + \frac{3\Delta h}{h} + \frac{\Delta\delta}{\delta} \ , \tag{A4}$$

where $E$ is sea ice strain modulus, and $\delta$ is the midspan deflection of beam. $\Delta\delta/\delta = 1$ % according to the performance of
displacement sensor, and the other parameters are taken the same values in Eqs. (A1) and (A2). Therefore, $\Delta E/E$ is 2.1 %.

For sea ice uniaxial compressive strength, it is calculated based on measurements of failure load and sample dimensions (see
Eq. 3). Uncertainty of sea ice uniaxial compressive strength is given by Eqs. (A5) and (A6):

$$\frac{\Delta\sigma_{\mathrm{c}}}{\sigma_{\mathrm{c}}} = \left|\frac{\partial\ln\sigma_{\mathrm{c}}}{\partial F_{\mathrm{max}}}\right|\Delta F_{\mathrm{max}} + \left|\frac{\partial\ln\sigma_{\mathrm{c}}}{\partial b}\right|\Delta b + \left|\frac{\partial\ln\sigma_{\mathrm{c}}}{\partial d}\right|\Delta d \ , \tag{A5}$$

$$\frac{\Delta\sigma_{\mathrm{c}}}{\sigma_{\mathrm{c}}} = \frac{\Delta F_{\mathrm{max}}}{F_{\mathrm{max}}} + \frac{\Delta b}{b} + \frac{\Delta d}{d} \ , \tag{A6}$$

where $\sigma_{\mathrm{c}}$ is sea ice uniaxial compressive strength, $F_{\mathrm{max}}$ is maximum recorded force, and $d$ is sample length. $\Delta F/F = 0.5$ %
according to the performance of force sensor. $\Delta b = \Delta d = 0.02$ mm. $b = 68.5$ mm and $d = 68.2$ mm taking the average sample
dimensions. Therefore, $\Delta\sigma_{\mathrm{c}}/\sigma_{\mathrm{c}}$ is 0.6 %.

Sea ice density is determined using mass/volume method (Eq. A7), and the uncertainty of sea ice density is given by Eqs. (A8)
and (A9):

$$\rho = \frac{M}{bhd} \ , \tag{A7}$$



$$\frac{\Delta\rho}{\rho} = \left|\frac{\partial \ln \rho}{\partial M}\right|\Delta M + \left|\frac{\partial \ln \rho}{\partial b}\right|\Delta b + \left|\frac{\partial \ln \rho}{\partial h}\right|\Delta h + \left|\frac{\partial \ln \rho}{\partial d}\right|\Delta d \, , \tag{A8}$$

$$\frac{\Delta\rho}{\rho} = \frac{\Delta M}{M} + \frac{\Delta b}{b} + \frac{\Delta h}{h} + \frac{\Delta d}{d} \, , \tag{A9}$$

where $M$ is the mass of sea ice samples, and $\rho$ is sea ice density.

For samples in the flexural tests, $b$ = 68.3 mm, $h$ = 67.7 mm, and $d$ = 550.1 mm taking the average sample dimensions. $M$ = 2033.5 g taking the average sample mass. $\Delta M$ = 0.1 g, $\Delta b = \Delta h$ = 0.02 mm, $\Delta d$ = 1 mm. Therefore, $\Delta\rho/\rho$ for ice samples in the flexural tests is 0.2 %.

For samples in the uniaxial compressive strength tests, $b$ = 68.5 mm, $d$ = 68.2 mm, and $h$ = 173.2 mm taking the average sample dimensions. $M$ = 624.5 g taking the average sample mass. $\Delta M$ = 0.1 g, $\Delta b = \Delta d$ = 0.02 mm, $\Delta h$ = 1 mm. Therefore, $\Delta\rho/\rho$ for ice samples in the flexural tests is 0.7 %.

## Data availability

All data are available at the website https://doi.org/10.5281/zenodo.7536404 (Wang et al., 2023).

## Author contribution

QW planned the experiments, and YL carried them out. QW analyzed the data and wrote the manuscript with contributions from all co-authors. ZL and PL reviewed and edited the manuscript.

## Competing interests

The authors declare that they have no conflict of interest.

## Acknowledgements

This research has been supported by the National Natural Science Foundation of China (grant nos. 42276242 and 52192692), and the Fundamental Research Funds for the Central Universities (grant no. DUT21RC3086).

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
