# Peer review of "The porosity effect on the mechanical properties of summer sea ice in the Arctic"

_The Cryosphere, 2023_

## Referee Comment (RC3)

[referee-annotated manuscript omitted]

---

## Referee Comment (RC4)

[referee-annotated manuscript omitted]

---

## Author Comment (AC2)

We appreciate warmly for the reviewer's earnest work. The comment is constructive, and we will revise the manuscript accordingly. Detailed answer to the comment is provided below.

**Comment:** Investigation on ice properties is valuable to the assessment of ship navigation in ice-covered waters. This work is quite important and practical to engineering application. The manuscript is well written and suggested to publish. I have a comment on Fig.12. It shows that the measured data are a little scattered from the varying trend line. More explanations are expected to improve confidence level of existing data.

**Response:**

(1) We have checked the confidence levels of the fitted varying trends, and the results showed that they are not significant at 0.1 level. So, we will revise the original figure as below and delete the varying trends.

[Figure]

(2) It could be seen from the revised figure that the strength of both first-year and multiyear ice decreased in the summer Arctic in recent years. Especially after the year 1990, the declining rates of both flexural strength and uniaxial compressive strength of multiyear ice were faster than those before the year 1990. Admittedly, the calculated sea ice strength was derived from sea ice porosities measured in only several different regions in the Arctic Ocean, and the data was a little scattered. More field measurements on Arctic sea ice physical and mechanical properties are still required to reveal their response to global warming.

We will give the above explanation in the revised manuscript.